# Improving Differentiable Neural Computers Through Memory Masking, De-allocation, and Link Distribution Sharpness Control

**Róbert Csordás**
The Swiss AI Lab, IDSIA / USI / SUPSI
`robert@idsia.ch`

**Jürgen Schmidhuber**
The Swiss AI Lab, IDSIA / USI / SUPSI
NNAISENSE
`juergen@idsia.ch`

## Abstract

The Differentiable Neural Computer (DNC) can learn algorithmic and question answering tasks. An analysis of its internal activation patterns reveals three problems: Most importantly, the lack of *key-value* separation makes the address distribution resulting from content-based look-up noisy and flat, since the *value* influences the score calculation, although only the *key* should. Second, DNC's de-allocation of memory results in aliasing, which is a problem for content-based look-up. Thirdly, chaining memory reads with the temporal linkage matrix exponentially degrades the quality of the address distribution. Our proposed fixes of these problems yield improved performance on arithmetic tasks, and also improve the mean error rate on the bAbI question answering dataset by $43\%$.

## 1 Introduction

Although Recurrent Neural Networks (RNNs) such as LSTM (Hochreiter & Schmidhuber, 1997; Gers et al., 2000) are in theory capable of solving complex algorithmic tasks (Siegelmann & Sontag, 1992), in practice they often struggle to do so. One reason is the large amount of time-varying memory required for many algorithmic tasks, combined with quadratic growth of the number of trainable parameters of a fully connected RNN when increasing the size of its internal state. Researchers have tried to address this problem by incorporating an *external memory* as a useful architectural bias for algorithm learning (Das et al., 1992; Mozer & Das, 1993; Graves et al., 2014; 2016).

Especially the Differentiable Neural Computer (DNC; Graves et al. (2016)) has shown great promise on a variety of algorithmic tasks – see diverse experiments in previous work (Graves et al., 2016; Rae et al., 2016). It combines a large external memory with advanced addressing mechanisms such as content-based look-up and temporal linking of memory cells. Unlike approaches that achieve state of the art performance on specific tasks, e.g. MemNN (Sukhbaatar et al., 2015) or Key-Value Networks (Miller et al., 2016) for the bAbI dataset (Weston et al., 2015), the DNC consistently reaches near state of the art performance on all of them. This generality makes the DNC worth of further study.

Three problems with the current DNC revolve around the *content-based look-up mechanism*, which is the main memory addressing system, and the *temporal linking* used to read memory cells in the same order in which they were written. First, the lack of key-value separation negatively impacts the accuracy of content retrieval. Second, the current de-allocation mechanism fails to remove de-allocated data from memory, which prevents the network from erasing outdated information without explicitly overwriting the data. Third, with each write, the noise from the write address distribution accumulates in the temporal linking matrix, degrading the overall quality of temporal links.

Here we propose a solution to each of these problems. We allow for dynamic key-value separation through a masking of both look-up key and data that is more general than a naive fixed key-value memory, yet does not suffer from loss of accuracy in addressing content. We propose to wipe the content of a memory cell in response to a decrease of its usage counter to allow for proper memory de-allocation. Finally, we reduce the effect of noise accumulation in the temporal linking matrix

through exponentiation and re-normalization of the links, resulting in improved sharpness of the corresponding address distribution.

These improvements are orthogonal to other previously proposed DNC modifications. Incorporation of the differentiable allocation mechanism Ben-Ari & Bekker (2017) or certain improvements to memory usage and computational complexity (Rae et al., 2016) might further improve the results reported in this paper. Certain bAbI-specific modifications Franke et al. (2018) are also orthogonal to our work.

We evaluate each of the proposed modifications empirically on a benchmark of algorithmic tasks and on bAbI (Weston et al., 2015). In all cases we find that our model outperforms the DNC. In particular, on the bAbI task we observe a $43\%$ relative improvement in terms of mean error rate. We find that improved de-allocation together with sharpness enhancement leads to zero error and 3x faster convergence on the large repeated copy task, while DNC is not able to solve it at all.

Section 2 provides a brief overview of the DNC. Section 3 discusses identified problems and proposed solutions in more detail. Section 4 analyzes these modifications one-by-one, demonstrating their positive effects.

## 2 DIFFERENTIBLE NEURAL COMPUTER

Here we provide a brief overview of the Differentiable Neural Computer (DNC). More details can be found in the original work of Graves et al. (2016).

The DNC combines a neural network (called controller) with an external memory that includes several supporting modules (subsystems) to do: read and write memory, allocate new memory cells, chain memory reads in the order in which they were written, and search memory for partial data. A simplified block diagram of the memory access is shown in Fig. 1.

**External memory**    A main component is the external fixed 2D memory organized in cells ($M_t \in \mathbb{R}^{N \times W}$, where N is the number of cells, W is the cell length). $N$ is independent of the number of trainable parameters. The controller is responsible for producing the activations of gates and keys controlling the memory transactions. The memory is accessed through multiple read heads and a single write head. Cells are addressed through a distribution over the whole address space. Each cell is read from and written to at every time step as determined by the address distributions, resulting in a differentiable procedure.

**Memory addressing**    The DNC uses three addressing methods. The most important one is content-based look-up. It compares every cell to a key ($\mathbf{k}_t^* \in \mathbb{R}^W$) produced by the controller, resulting in a score, which is later normalized to get an address distribution over the whole memory. The second is the temporal linking, which has 2 types: forward and backward. They show which cell is being written after and before the one read in the previous time step. They are useful for processing sequences of data. A so-called temporal linkage matrix ($L_t \in \mathbb{R}^{N \times N}$) is used to project any address distribution to a distribution that follows ($\mathbf{f}_t^i \in [0, 1]^N$) or precedes it ($\mathbf{b}_t^i \in [0, 1]^N$). The third is the allocation mechanism, which is used only for write heads, and used when a new memory cell is required.

**Memory allocation**    Memory allocation works by maintaining usage counters for every cell. These are incremented on memory writes and optionally decremented on memory reads (de-allocation). When a new cell is allocated, the one with the lowest usage counter is chosen. De-allocation is controlled by a gate, which is based on the address distribution of the previous read and decreases the usage counter of each cell.

**Read / Write**    The memory is first written to, then read from. A write address is generated as a weighted average of the write content-based look-up and the allocation distribution. The update is done in a gated way by erasing vector $\mathbf{e}_t \in [0, 1]^W$. Parallel to the write, the temporal linkage matrix is also updated. Finally the memory is read from. The final read address distribution ($\mathbf{w}_t^{r,i} \in [0, 1]^N$) is generated as the weighted average of the read content-based look-up distribution and forward and backward temporal links. Memory cells are averaged based on this address, resulting in a single

vector, which is the retrieved data. This data is combined with the output of the controller to produce the model's final output.

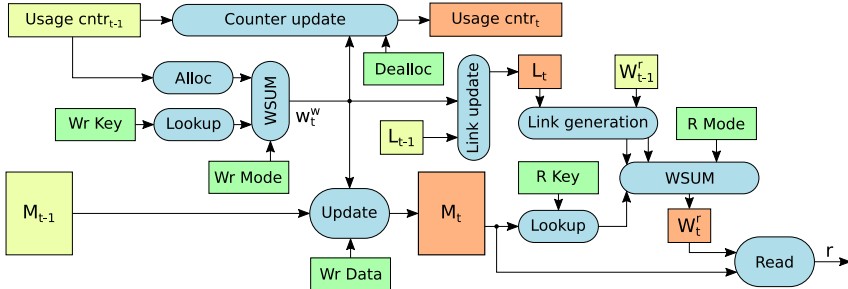

Figure 1: Simplified block diagram of DNC's memory access module with single read head. Yellow boxes denote the inputs from the previous time step, orange boxes are the corresponding outputs to the next time step. Green boxes are the control inputs from the controller. Blue, rounded boxes are modules responsible for a specific function. $\mathbf{w}_t^w$ denotes the write address, $\mathbf{w}_t^r$ the read address, $\boldsymbol{L}_t$ the temporal linkage matrix. $\boldsymbol{M}_t$ is the memory. Arrow "r" denotes the output of the memory read.

## 3 METHOD

### 3.1 MASKED CONTENT-BASED ADDRESSING

The goal of content-based addressing is to find memory cells similar to a given key. The query key contains partial information (it is a partial memory), and the content-based memory read completes its missing (unknown) part based on previous memories. However, controlling which part of the key vector to search for is difficult because there is no key-value separation: the entire key and entire cell value are compared to produce the similarity score. This means that the part of the cell value that is unknown during search time and should be retrieved is also used in the normalization part of the cosine similarity, resulting in an unpredictable score. The shorter the known part and the longer the part to be retrieved, the worse the problem. This might result in less similar cells having higher scores, and might make the resulting address distribution flat because of the division by the length of the data before the softmax acts as an increased temperature parameter. Imagine a situation when the network has to tag some of the data written into the memory (for example because it is the start of the sequence in the repeated copy task, see Section 4). For tagging it can use only a single element of the memory cell, using the rest for the data that has to be stored. Later, when the network has to find this tag, it searches for it, specifying only that single element of the key. But the resulting score is also normalized by the data that is stored along with the tag, which takes most of the memory cell, hence the resulting score will almost completely be dominated by it.

The problem can be solved by providing a way of explicitly masking the part that is unknown and should not be used in the query. This is more general than key-value memory, since the key-value separation can be controlled dynamically and does not suffer from the incorrect score problem. We achieve this by producing a separate mask vector $\mathbf{m}_t^* \in [0-1]^W$ through the controller, and multiplying both the search key and the memory content by it before comparing ($\beta$ is the write key strength controlling the temperature of the softmax):

$$C(\boldsymbol{M}, \mathbf{k}, \beta, \mathbf{m})[i] = softmax(D\left(\mathbf{k} \odot \mathbf{m}, \boldsymbol{M} \odot \mathbf{1}\mathbf{m}^T\right)\beta)$$

$$\mathbf{c}_t^w = C\left(\boldsymbol{M}_{t-1}, \mathbf{k}_t^w, \beta_t^w, \mathbf{m}_t^w\right) \qquad c_t^{r,i} = C\left(\boldsymbol{M}_t, \mathbf{k}_t^{r,i}, \beta_t^{r,i}, \mathbf{m}_t^{r,i}\right)$$

Fig. 2 shows how the masking step is incorporated in the address generation of the DNC.

### 3.2 DE-ALLOCATION AND CONTENT-BASED LOOK-UP

The DNC tracks allocation states of memory cells by so-called usage counters which are increased on memory writes and optionally decreased after reads. When allocating memory, the cell with the lowest usage is chosen. Decreasing is done by element-wise multiplication with a so-called retention

vector ($\psi_t$), which is a function of previously read address distributions ($\mathbf{w}_{t-1}^{r,i}$) and scalar gates. Vector $\psi_t$ indicates how much of the current memory should be kept. The problem is that it affects solely the usage counters and not the actual memory $\boldsymbol{M}_t$. But memory content plays a vital role in both read and write address generation: the content based look-up still finds de-allocated cells, resulting in memory aliasing. Consider the repeated copy task (Section 4), which needs repetitive allocation during storage of a sequence and de-allocation after it was read. The network has to store and repeat sequences multiple times. It has to tag the sequence beginning to know from where the repetition should start from. This could be done by content-based look-up. During the repetition phase, each cell read is also de-allocated. However when the repetition of the second sequence starts, the search for the tagged cell can find both the old and the new marked cell with equal score, making it impossible to determine which one is the correct match. We propose to zero out the memory contents by multiplying every cell of the memory matrix $\boldsymbol{M}_t$ with the corresponding element of the retention vector. Then the memory update equation becomes:

$$\boldsymbol{M}_t = \boldsymbol{M}_{t-1} \odot \psi_t \mathbf{1}^T \odot (\boldsymbol{E} - \mathbf{w}_t^w \mathbf{e_t}^\mathsf{T}) + \mathbf{w}_t^w \mathbf{v}_t^\mathsf{T} \qquad (1)$$

where $\odot$ is the element-wise product, $\mathbf{1} \in \mathbb{R}^N$ is a vector of ones, $\mathbf{E} \in \mathbb{R}^{N \times W}$ is a matrix of ones. Note that the cosine similarity (used for comparing the key to the memory content) is normalized by the length of the memory content vector which would normally cancel the effect of Eq. 1. However, in practice, due to numerical stability, cosine similarity is implemented as

$$D(\mathbf{u}, \mathbf{v}) = \frac{\mathbf{u} \cdot \mathbf{v}}{|\mathbf{u}||\mathbf{v}| + \epsilon}$$

where $\epsilon$ is a small constant. In practice, free gates $f_t^i$ tend to be almost 1, so $\psi_t$ is very close to 0, making the stabilizing constant $\epsilon$ dominant with respect to the norm of the erased memory content vector. This will assign a low score to the erased cell in the content addressing: the memory is totally removed.

## 3.3 SHARPNESS OF TEMPORAL LINK DISTRIBUTIONS

With temporal linking, the model is able to sequentially read memory cells in the same or reverse order as they were written. For example, repeating a sequence is possible without content-based look-up: the forward links $\mathbf{f}_t^i$ can be used to jump to the next cell. Any address distribution can be projected to the next or the previous one through multiplying it by a so-called temporal link matrix ($\boldsymbol{L}_t$) or its transpose. $\boldsymbol{L}_t$ can be understood as a continuous adjacency matrix. On every write all elements of $\boldsymbol{L}_t$ are updated to an extent controlled by the write address distribution ($\mathbf{w}_t^w$). Links related to previous writes are weakened; the new links are strengthened. If $\mathbf{w}_t^w$ is not one-hot, sequence information about all non-zero addresses will be reduced in $\boldsymbol{L}_t$ and the noise from the current write will also be included repeatedly. This will make forward ($\mathbf{f}_t^i$) and backward ($\mathbf{b}_t^i$) distributions of long-term-present cells noisier and noisier, and flatten them out. When chaining multiple reads by temporal links, the new address is generated through repeatedly multiplying by $\boldsymbol{L}_t$, making the blurring effect exponentially worse.

We propose to add the ability to improve sharpness of the link distribution ($\mathbf{f}_t^i$ and $\mathbf{b}_t^i$). This does not fix the noise accumulation in the link matrix $\boldsymbol{L}_t$, but it significantly reduces the effect of exponential blurring behavior when following the temporal links, making the noise in $\boldsymbol{L}_t$ less harmful. We propose to add an additional sharpness enhancement step $S(\mathbf{d}, s)_i$ to the temporal link distribution generation. By exponentiating and re-normalizing the distribution, the network is able to adaptively control the importance of non-dominant elements of the distribution.

$$\mathbf{f}_t^i = S\left(\boldsymbol{L}_t \mathbf{w}_{t-1}^{r,i}, s_t^{f,i}\right) \qquad \mathbf{b}_t^i = S\left(\boldsymbol{L}_t^\mathsf{T} \mathbf{w}_{t-1}^{r,i}, s_t^{b,i}\right) \qquad S(\mathbf{d}, s)_i = \frac{(\mathbf{d}_i)^s}{\sum_j (\mathbf{d}_j)^s} \qquad (2)$$

Scalars $s_t^{f,i} \in \mathbb{R}$ and $s_t^{b,i} \in \mathbb{R}$ should be generated by the controller ($\hat{s}_t^{f,i}$ and $\hat{s}_t^{b,i}$). The oneplus nonlinearity is used to bound them in range $[0, \infty)$: $s_t^{f,i} = oneplus(\hat{s}_t^{f,i})$ and $s_t^{b,i} = oneplus(\hat{s}_t^{b,i})$. Note that $S(\mathbf{d}, s)_i$ in Eq. 2 can be numerically unstable. We propose to stabilize it by:

$$S(\mathbf{d}, s)_i = \frac{\left(\frac{\mathbf{d}_i + \epsilon}{\max(\mathbf{d} + \epsilon)}\right)^s}{\sum_j \left(\frac{\mathbf{d}_j + \epsilon}{\max(\mathbf{d} + \epsilon)}\right)^s}$$

Fig. 2 shows the block diagram of read address generation in DNC with key masking and sharpness enhancement. The sharpness enhancement block is inserted into the forward and backward link generation path right before combining them with the content-based look-up distribution.

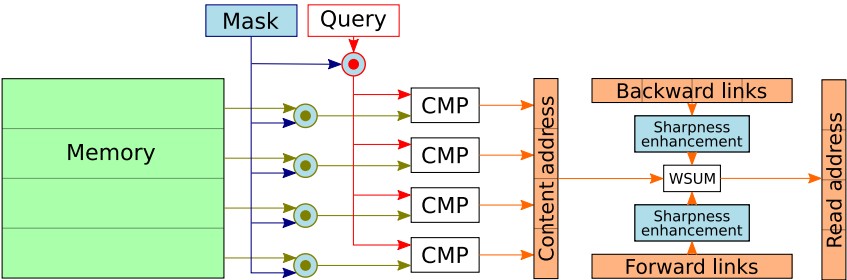

Figure 2: Block diagram of read address generation in DNC with key masking and sharpness enhancement. Blue parts indicate new components absent in standard DNC. CMP is a cosine similarity-based comparator. Memory and key are compared after a novel masking step. Before combining temporal links and content-based address distribution, sharpness enhancement takes place.

## 4 EXPERIMENTS

To analyze the effects of our modifications we used simple synthetic tasks designed to require most DNC parts while leaving the internal dynamics somewhat human interpretable. The tasks allow for focusing on the individual contributions of specific network components. We also conducted experiments on the much more complex bAbI dataset (Weston et al., 2015).

We tested several variants of our model. For clarity, we use the following notation: DNC is the original network Graves et al. (2016), DNC-D has modified de-allocation, DNC-S has added sharpness enhancement, DNC-M has added masking in content based addressing. Multiple modifications (D, M, S) can be present.

**Copy Task** A sequence of length $L$ of binary random vectors of length $W$ is presented to the network, and the network is required to repeat them. The repeat phase starts with a special input token, after all inputs are presented. To solve this task the network has to remember the sequence, which requires allocating and recalling from memory. However, it does not require memory de-allocation and reuse. To force the network to demonstrate its de-allocation capabilities, $N$ instances of such data are generated and concatenated. Because the total length of the $N$ sequences exceeds the number of cells in memory, the network is forced to reuse its memory cells. An example is shown in Fig. 4a.

**Associative Recall Task** In the associative recall task (Graves et al. (2014)) $B$ blocks of $W_b$ words of length $W$ are presented to the network sequentially, with special bits indicating the start of each block. After presenting the input to the network, a special bit indicates the start of the recall phase where a randomly chosen block is repeated. The network needs to output the next block in the sequence.

**Key-Value Retrieval Task** The key-value retrieval task demonstrates some properties of memory masking. $L$ words of length $2W$ are presented to the network. Words are divided in two parts of equal length, $W_1$ and $W_2$. All the words are presented to the network. Next the words are shuffled, parts $W_1$ are fed to the network, requiring it to output the missing part $W_2$ for every $W_1$. Next, the words are shuffled again, $W_2$ is presented and the corresponding $W_1$ is requested. The network must be able to query its memory using either part of the words to complete this task.

### 4.1 IMPLEMENTATION DETAILS

Our PyTorch implementation is available on `https://github.com/xdever/dnc`. We provide equations for our DNC-DMS model in Appendix A. Following Graves et al. (2016), we trained

all networks using RMSProp (Tieleman & Hinton (2012)), with a learning rate of $10^{-4}$, momentum 0.9, $\epsilon = 10^{-10}$, $\alpha = 0.99$. All parameters except the word embedding vectors and biases have a weight decay of $10^{-5}$. For task-specific hyperparameters, check Appendix B.

## 4.2 THE EFFECT OF MODIFICATIONS

**Masking**  Fig. 3a shows the performance of various models on the associative recall task. The two best performing models use memory masking. From the standard deviation it can be seen that many seeds converge much faster with masking than without. Sharpening negatively impacts performance on this task (see Section 4.3 for further discussion). Note that $\delta > 0$ (we used $\delta = 0.1$) is essential for good performance, while $\delta = 0$ slows down convergence speed (see Equation 4 in Appendix A).

To demonstrate that the system learns to use masking dynamically instead of learning a static weighting, we trained DNC-M on the key-value retrieval task. Fig. 3b shows how the network changes the mask when the query switches from $W_1$ to $W_2$. Parts of the mask activated during the $W_1$ period almost complement the part activated during the $W_2$ period, just like the query keys do.

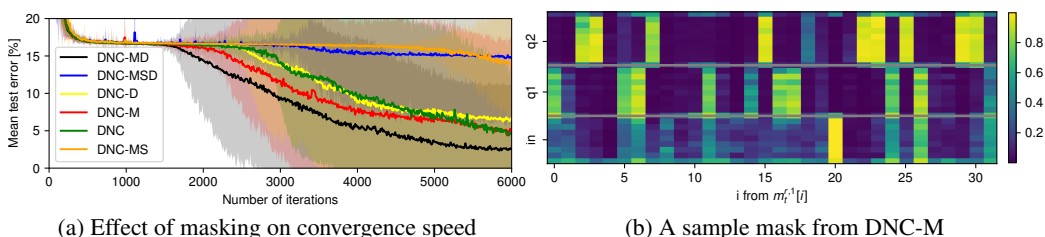

(a) Effect of masking on convergence speed     (b) A sample mask from DNC-M

Figure 3: (a) Mean training loss on the associative recall task. The shaded area shows the $\pm 2\sigma$ mark (12 seeds/model). Masking improves convergence speed. (b) An example read mask of DNC-M in the key-value retrieval task. Yellow values indicate parts of the key the network searches for, the blue values indicate parts that need to be retrieved form memory. When the query switches from $W_1$ to $W_2$, the mask changes. In the bottom third (in) the input is stored (look-up is not used). For in middle third (q1) $W_1$ is presented in random order and $W_2$ is retrieved. In the last third (q2) $W_2$ is presented in random order and $W_1$ is retrieved.

**De-allocation**  Graves et al. (2016) successfully trained DNC on the repeat copy task with a small number of repeats ($N$) and relatively short length ($L$). We found that increasing $N$ makes DNC fail to solve the task (see Fig. 4a). Fig. 4b shows that our model solves the task perfectly. Its outputs are clean and sharp. Furthermore it converges much faster than DNC, reaching near-zero loss very quickly. We hypothesize that the reason for this is the modified de-allocation: the network can store the beginning of every sequence with similar key without causing look-up conflicts, as it is guaranteed that the previously present key is wiped from memory. DNC seems able to solve the short version of the problem by learning to use different keys for every repeat step, which is not a general solution. This hypothesis, however, is difficult to prove, as neither the write vector nor the look-up key is easily human-interpretable.

**Sharpness enhancement**  To analyze the problem of degradation of temporal links after successive link matrix updates, we examined the forward and backward link distributions ($\mathbf{f}_t^i$ and $\mathbf{b}_t^i$) of the model with modified deallocation (DNC-D). The forward distribution is shown in Fig. 5a. The problem presented in Section 3.3 is clearly visible: the distribution is blurry; the problem becomes worse with each iteration. Fig. 5c shows the read mode ($\pi_t^1$) of the same run. It is clear that only content based addressing (middle column) is used. When the network starts repeating a block, the weight of forward links (last column) is increased a bit, but as the distribution becomes more blurred, a pure content-based look-up is preferred. Probably it is easier for the network to perform a content-based look-up with a learned counter as a key rather than to learn to restore the corrupted data from blurry reads. Fig. 5b shows the forward distributions of the model with sharpness enhancement as suggested in Section 3.3 for the same input. The distribution is much sharper, staying sharp until the

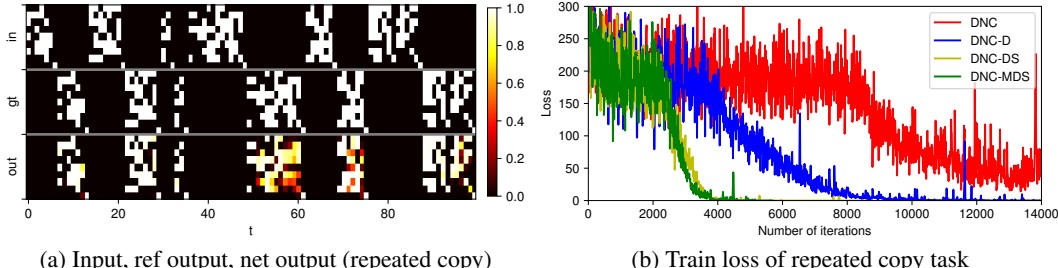

(a) Input, ref output, net output (repeated copy)    (b) Train loss of repeated copy task

Figure 4: (a) Input (top), ground truth (middle), and network output (bottom) of DNC on big repeat copy tasks. DNC fails to solve the task; the output is blurry. The problem is especially apparent starting from $t = 50$. (b) De-allocating and sharpness enhancement substantially improves convergence speed. The improvement by the masking is marginal, probably because the task uses temporal links.

very end of the repeat block. The read mode ($\pi_t^1$) for the same run can be seen in Fig. 5d. Obviously the network prefers to use the links in this case.

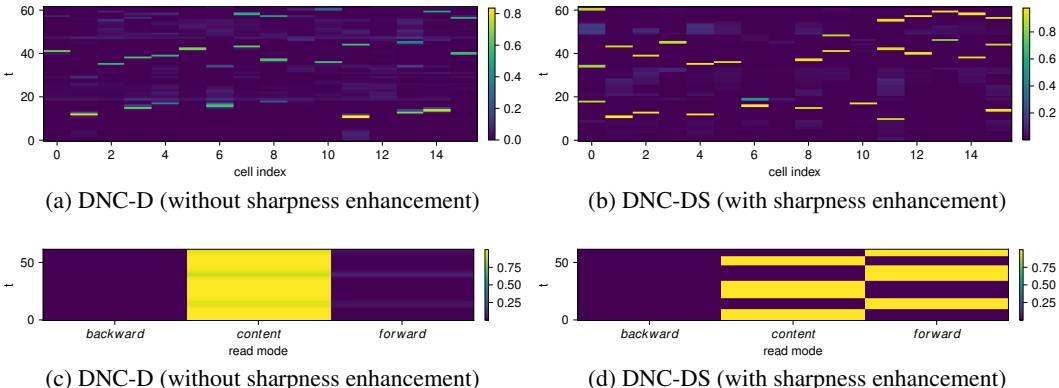

(a) DNC-D (without sharpness enhancement)    (b) DNC-DS (with sharpness enhancement)

(c) DNC-D (without sharpness enhancement)    (d) DNC-DS (with sharpness enhancement)

Figure 5: (a), (b) Example forward link distribution. Each row is an address distribution across all memory cells. Blue cells are not read, yellow cells are read with a large weight. (a) DNC-D: without sharpness enhancement the distributions are blurred, rarely having peaks near 1.0. The problem becomes worse over time. 3 repeats are shown. Notice the more intense blocks for $t \in [9, 18], [34, 46]$ and $[54, 62]$. (b) Sharpness enhancement (DNC-DS) makes the distribution sharp during the read, peaking near 1.0. Note that (a) and (b) have identical input data. (c), (d) The $\pi_t^1$ distribution for (a) and (b). Columns are the weighting of the backward links, the content based look up, and the forward links, respectively. (c) The forward links are barely used without sharpness enhancement. (d) With sharpness enhancement the forward links are used for every block.

### 4.3 BABI EXPERIMENTS

bAbI (Weston et al. (2015)) is an algorithmically generated question answering dataset containing 20 different tasks. Data is organized in sequences of sentences called stories. The network receives the story word by word. When a question mark is encountered, the network must output a single word representing the answer. A task is considered solved if the error rate (number of correctly predicted answer words divided by the number of total predictions) of the network decreases below $5\%$, as usual for this task.

Manually analyzing bAbI tasks let us to believe that some are difficult to solve within a single timestep. Consider the sample from QA16: "Lily is a swan. Bernhard is a lion. Greg is a swan. Bernhard is white. Brian is a lion. Lily is gray. Julius is a rhino. Julius is gray. Greg is gray. What color

Table 1: bAbI error rates of different models after 0.5M iterations of training [%]

| Task | DNC (ours) | DNC-MDS | DNC-DS | DNC-MS | DNC-MD | Graves et al |
|------|-----------|---------|--------|--------|--------|--------------|
| 1 | $2.5 \pm 4.4$ | $0.4 \pm 1.2$ | $0.7 \pm 1.6$ | $0.0 \pm 0.1$ | $\mathbf{0.0 \pm 0.0}$ | $9.0 \pm 12.6$ |
| 2 | $29.0 \pm 19.4$ | $8.6 \pm 10.1$ | $18.6 \pm 15.1$ | $7.8 \pm 5.9$ | $\mathbf{6.9 \pm 4.7}$ | $39.2 \pm 20.5$ |
| 3 | $32.3 \pm 14.7$ | $10.8 \pm 9.5$ | $16.9 \pm 13.0$ | $\mathbf{7.9 \pm 7.8}$ | $12.4 \pm 5.1$ | $39.6 \pm 16.4$ |
| 4 | $\mathbf{0.8 \pm 1.5}$ | $0.8 \pm 1.5$ | $6.4 \pm 10.0$ | $0.8 \pm 1.0$ | $0.1 \pm 0.2$ | $0.4 \pm 0.7$ |
| 5 | $1.5 \pm 0.6$ | $1.6 \pm 1.0$ | $\mathbf{1.3 \pm 0.5}$ | $1.7 \pm 1.1$ | $1.3 \pm 0.7$ | $1.5 \pm 1.0$ |
| 6 | $5.2 \pm 6.8$ | $1.1 \pm 2.1$ | $2.4 \pm 3.8$ | $\mathbf{0.0 \pm 0.1}$ | $0.1 \pm 0.1$ | $6.9 \pm 7.5$ |
| 7 | $8.8 \pm 5.8$ | $3.4 \pm 2.3$ | $7.6 \pm 5.1$ | $\mathbf{2.5 \pm 2.0}$ | $3.0 \pm 5.0$ | $9.8 \pm 7.0$ |
| 8 | $11.6 \pm 9.4$ | $4.6 \pm 4.5$ | $10.9 \pm 7.9$ | $\mathbf{1.8 \pm 1.6}$ | $2.5 \pm 2.1$ | $5.5 \pm 5.9$ |
| 9 | $4.5 \pm 5.8$ | $0.8 \pm 1.9$ | $2.0 \pm 3.3$ | $\mathbf{0.1 \pm 0.2}$ | $0.1 \pm 0.2$ | $7.7 \pm 8.3$ |
| 10 | $9.1 \pm 11.5$ | $2.6 \pm 3.9$ | $4.1 \pm 5.9$ | $0.6 \pm 0.6$ | $\mathbf{0.5 \pm 0.5}$ | $9.6 \pm 11.4$ |
| 11 | $11.6 \pm 9.4$ | $0.1 \pm 0.1$ | $0.1 \pm 0.2$ | $\mathbf{0.0 \pm 0.0}$ | $\mathbf{0.0 \pm 0.0}$ | $3.3 \pm 5.7$ |
| 12 | $1.1 \pm 0.8$ | $\mathbf{0.2 \pm 0.2}$ | $0.5 \pm 0.4$ | $0.3 \pm 0.4$ | $\mathbf{0.2 \pm 0.2}$ | $5.0 \pm 6.3$ |
| 13 | $1.1 \pm 0.8$ | $\mathbf{0.1 \pm 0.1}$ | $0.2 \pm 0.2$ | $0.2 \pm 0.2$ | $\mathbf{0.1 \pm 0.1}$ | $3.1 \pm 3.6$ |
| 14 | $24.8 \pm 22.5$ | $8.0 \pm 13.1$ | $20.0 \pm 19.4$ | $1.8 \pm 0.9$ | $\mathbf{2.0 \pm 1.6}$ | $11.0 \pm 7.5$ |
| 15 | $40.8 \pm 1.4$ | $26.3 \pm 20.7$ | $42.1 \pm 6.3$ | $33.0 \pm 15.1$ | $\mathbf{23.6 \pm 18.6}$ | $27.2 \pm 20.1$ |
| 16 | $\mathbf{53.1 \pm 1.2}$ | $54.5 \pm 1.8$ | $53.5 \pm 1.4$ | $53.2 \pm 2.3$ | $53.9 \pm 1.2$ | $53.6 \pm 1.9$ |
| 17 | $\mathbf{37.8 \pm 2.5}$ | $39.9 \pm 3.2$ | $40.1 \pm 2.0$ | $41.2 \pm 3.0$ | $39.8 \pm 1.2$ | $32.4 \pm 8.0$ |
| 18 | $7.0 \pm 3.0$ | $6.3 \pm 4.1$ | $9.4 \pm 0.9$ | $3.3 \pm 2.2$ | $\mathbf{2.0 \pm 2.6}$ | $4.2 \pm 1.8$ |
| 19 | $67.6 \pm 8.6$ | $48.6 \pm 32.8$ | $67.6 \pm 7.9$ | $48.1 \pm 26.7$ | $\mathbf{40.7 \pm 34.9}$ | $64.6 \pm 37.4$ |
| 20 | $\mathbf{0.0 \pm 0.0}$ | $0.9 \pm 0.9$ | $1.5 \pm 1.0$ | $5.3 \pm 12.5$ | $0.1 \pm 0.1$ | $0.0 \pm 0.1$ |
| mean | $16.9 \pm 5.2$ | $11.0 \pm 3.8$ | $15.3 \pm 3.5$ | $10.5 \pm 1.9$ | $\mathbf{9.5 \pm 1.6}$ | $16.7 \pm 7.6$ |

is Brian? *A: white*" The network should be able to "think for a while" about the answer: it needs to do multiple memory searches to chain the clues together. This cannot be done in parallel as the result of one query is needed to produce the key for the next. One solution would be to use adaptive computation time (Schmidhuber (2012), Graves (2016)). However, that would add an extra level of complexity to the network. So we decided to insert a constant $T = 3$ blank steps before every answer of the network—a difference to what was done previously Graves et al. (2016). We also use a word embedding layer, instead of one-hot input representation, as is typical for NLP tasks. The embedding layer is a learnable lookup-table that transforms word indices to a learnable vector of length $E$.

In Table 1, we present experimental results of multiple versions of the network after $0.5M$ iterations with batch size 2. The performance reported by Graves et al. (2016) is also shown (column Graves et al). Our best performing model (DNC-MD) reduces the mean error rate by $43\%$, while having also a lower variance. This model does not use sharpness enhancement, which penalizes mean performance by only $1.5\%$ absolute. We hypothesize this is due to the nature of the task, which rarely needs step-to-step transversal of words, but requires many content-based look-ups. When following a path of an object, many words and even sentences might be irrelevant between the clues, so the sequential linking in order of writing is of little to no use. Compare Franke et al. (2018) where the authors completely removed the temporal linking for bAbI. However, we argue that for other kinds of tasks the link distribution sharpening might be very important (see Fig. 4b, where sharpening helps and masking does not).

Mean test error curves are shown on Fig. 6 in Appendix C. Our models converge faster and have both lower error and lower variance than DNC. (Note that our goal was not to achieve state of the art performance (Santoro et al., 2017; Henaff et al., 2016; Dehghani et al., 2018) on bAbI. It was to exhibit and overcome certain shortcomings of DNC.)

## 5 CONCLUSION

We identified three drawbacks of the traditional DNC model, and proposed fixes for them. Two of them are related to content-based addressing: (1) Lack of key-value separation yields uncertain and noisy address distributions resulting from content-based look-up. We mitigate this problem by a

special masking method. (2) De-allocation results in memory aliasing. We fix this by erasing memory contents in parallel to decreasing usage counters. (3) We try to avoid the blurring of temporal linkage address distributions by sharpening the distributions.

We experimentally analyzed the effect of each novel modification on synthetic algorithmic tasks. Our models achieved convergence speed-ups on all them. In particular, modified de-allocation and masking in content-based look-up helped in every experiment we performed. The presence of sharpness enhancement should be treated as a hyperparameter, as it benefits some but not all tasks. Unlike DNC, DNC+MDS solves the large repeated copy task. DNC-MD improves the mean error rate on bAbI by $43\%$. The modifications are easy to implement, add only few trainable parameters, and hardly affect execution time.

In future work we'll investigate in more detail when sharpness enhancement helps and when it is harmful, and why. We also will investigate the possibility of merging our improvements with related work (Ben-Ari & Bekker, 2017; Rae et al., 2016), to further improve the DNC.

ACKNOWLEDGMENTS

The authors wish to thank Sjoerd van Steenkiste, Paulo Rauber and the anonymous reviewers for their constructive feedback. We are also grateful to NVIDIA Corporation for donating a DGX-1 as part of the Pioneers of AI Research Award and to IBM for donating a Minsky machine. This research was supported by an European Research Council Advanced Grant (no: 742870).

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

## A    IMPLEMENTATION DETAILS

Here we present the equations for our full model (DNC-MDS). The other models can be easily implemented in line with details of Section 3. We also highlight differences to the DNC by Graves et al. (2016).

The memory at step t is represented by matrix $\boldsymbol{M}_t \in \mathbb{R}^{N \times W}$, where N is the number of cells, W is the word length. The network receives an input $\mathbf{x}_t \in \mathbb{R}^X$ and produces output $\mathbf{y}_t \in \mathbb{R}^Y$. The controller of the network receives input vector $\mathbf{x}_t$ concatenated with all $R$ (number of read heads) read vectors $\mathbf{r}_{t-1}^1, ..., \mathbf{r}_{t-1}^R$ from the previous step, and produces output vector $\mathbf{h}_t$. The controller can be an LSTM or feedforward network, and may have single or multiple layers. The controller's output is mapped to the interface vector $\xi_t$ by matrix $\boldsymbol{W}_\xi \in \mathbb{R}^{2(W*R)+4W+7R+3}$ by $\xi_t = \boldsymbol{W}_\xi \mathbf{h}_t$. An immediate output vector $\mathbf{v}_t \in \mathbb{R}^Y$ is also generated: $\mathbf{v}_t = \boldsymbol{W}_y \mathbf{h}_t$. The output interface vector is split into many sub-vectors controlling various parts of the network:

$$\xi_t = [\mathbf{k}_t^{r,1}..\mathbf{k}_t^{r,R}; \hat{\beta}_t^{r,1}..\hat{\beta}_t^{r,R}; \mathbf{k}_t^w; \hat{\beta}_t^w; \hat{\mathbf{e}}_t; \mathbf{v}_t; \hat{f}_t^1..\hat{f}_t^R; \hat{g}_t^a; \hat{g}_t^w; \hat{\pi}_t^1..\hat{\pi}_t^R;$$
$$\hat{\mathbf{m}}_t^w; \hat{\mathbf{m}}_t^{r,1}..\hat{\mathbf{m}}_t^{r,R}, \hat{s}_t^{f,1}..\hat{s}_t^{f,R}, \hat{s}_t^{b,1}..\hat{s}_t^{b,R}] \quad (3)$$

Notation: $1 \leq i \leq R$ is the read head index; $\mathbf{k}_t^{r,i}$ are the keys used for read content-based address generation; $\beta_t^{r,i} = oneplus(\hat{\beta}_t^{r,i})$ are the read key strengths ($oneplus(x) = 1 + log(1 + e^x)$); $\mathbf{k}_t^w$ is the query key used for content-based address generation for writes; $\beta_t^w = oneplus(\hat{\beta}_t^w)$ is the write key strength; $\mathbf{e}_t = \sigma(\hat{e}_t)$ is the erase vector which acts as an in-cell gate for memory writes; $\mathbf{v}_t$ is the write vector which is the actual data being written; $f_t^i = \sigma(\hat{f}_t^i)$ are the free gates controlling whether to de-allocate the cells read in the previous step; $g_t^a = \sigma(\hat{g}_t^a)$ is the allocation gate; $g_t^w = \sigma(\hat{g}_t^w)$ is the write gate; $\pi_t^i = softmax(\hat{\pi}_t^i)$ are the read modes (controlling whether to use temporal links or content-based look-up distribution as read address); $s_t^{f,i} = oneplus(\hat{s}_t^{f,i})$ are the forward sharpness enhancement coefficients; $s_t^{b,i} = oneplus(\hat{s}_t^{b,i})$ are the backward sharpness enhancement coefficients.

Special care must be taken of the range of lookup masks $\mathbf{m}_t^w$ and $\mathbf{m}_t^{r,i}$. It must be limited to $(\delta, 1)$, where $\delta$ is a small real number. A $\delta$ close to 0 might harm gradient propagation by blocking gradients of masked parts of key and memory vector.

$$\mathbf{m}_t^w = \sigma(\hat{\mathbf{m}}_t^w) * (1 - \delta) + \delta \qquad \mathbf{m}_t^{r,i} = \sigma(\hat{\mathbf{m}}_t^{r,i}) * (1 - \delta) + \delta \quad (4)$$

We suggest initializing biases for $\hat{\mathbf{m}}_t^w$ and $\hat{\mathbf{m}}_t^{r,i}$ to 1 to avoid low initial gradient propagation.

Content-based look-up is used to generate an address distribution based on matching a key against memory content:

$$C(\boldsymbol{M}, \mathbf{k}, \beta, \mathbf{m})[i] = softmax(D(\mathbf{k} \odot \mathbf{m}, \boldsymbol{M} \odot \mathbf{1}\mathbf{m}^T)\beta) \quad (5)$$

Compare this to the $C(\boldsymbol{M}, \mathbf{k}, \beta, \mathbf{m})[i] = softmax(D(\mathbf{k}, \boldsymbol{M})\beta)$ of Graves et al. (2016).

Where $D$ is the row-wise cosine similarity with numerical stabilization:

$$D(\mathbf{u}, \boldsymbol{M})[i] = \frac{\mathbf{u} \cdot \boldsymbol{M}[i, \cdot]}{|\mathbf{u}||\boldsymbol{M}[i, \cdot]| + \epsilon} \quad (6)$$

The memory is first written to, then read from. To write the memory, allocation and content-based lookup distributions are needed. Allocation is calculated based on usage vectors $\mathbf{u}_t$. These are updated with the help of memory retention vector $\psi_t$:

$$\psi_t = \prod_{i=1}^R \left(\mathbf{1} - f_t^i \mathbf{w}_{t-1}^{r,i}\right) \quad (7)$$

$$\mathbf{u}_t = \left(\mathbf{u}_{t-1} + \mathbf{w}_{t-1}^w - \mathbf{u}_{t-1} \odot \mathbf{w}_{t-1}^w\right) \odot \psi_\mathbf{t}. \quad (8)$$

Operation $\odot$ is the element-wise multiplication. Free list $\phi_t$ is the list of indices of sorted memory locations in ascending order of their usage $\mathbf{u}_t$. So $\phi_t[1]$ is the index of the least used location. Then allocation address distribution $\mathbf{a_t}$ is

$$\mathbf{a}_t[\phi_t[j]] = (1 - \mathbf{u}_t[\phi_t[j]]) \prod_{i=1}^{j-1} \mathbf{u_t}[\phi_t[i]]$$

The write address distribution $w_t^w \in [0,1]^N$ is:

$$\mathbf{c}_t^w = C\left(\boldsymbol{M}_{t-1}, \mathbf{k}_t^w, \beta_t^w, \mathbf{m}_t^w\right) \tag{9}$$

$$\mathbf{w}_t = g_t^w \left[g_t^a \mathbf{a}_t + (1 - g_t^a)\mathbf{c}_t^w\right]$$

Memory is updated by ($\mathbf{1} \in \mathbb{R}^N$ is a vector of ones, $\boldsymbol{E} \in \mathbb{R}^{N \times W}$ is a matrix of ones):

$$\boldsymbol{M}_t = \boldsymbol{M}_{t-1} \odot \psi_t \mathbf{1}^T \odot \left(\boldsymbol{E} - \mathbf{w}_t^w \mathbf{e_t^{\mathsf{T}}}\right) + \mathbf{w}_t^w \mathbf{v}_t^{\mathsf{T}} \tag{10}$$

Compare this to the $\boldsymbol{M}_t = \boldsymbol{M}_{t-1} \odot \left(\boldsymbol{E} - \mathbf{w}_t^w \mathbf{e_t^{\mathsf{T}}}\right) + \mathbf{w}_t^w \mathbf{v}_t^{\mathsf{T}}$ of Graves et al. (2016).

To track the temporal distance of memory allocations, a temporal link matrix $\boldsymbol{L}_t \in [0,1]^{N \times N}$ is maintained. It is a continuous adjacency matrix. A helper quantity called precedence weighting is defined: $\mathbf{p}_0 = \mathbf{0}$ and

$$\mathbf{p}_t = \left(1 - \sum_i \mathbf{w}_t^w[i]\right) \mathbf{p}_{t-1} + \mathbf{w}_t^w$$

$$\boldsymbol{L}_0[i,j] = 0 \;\; \forall i,j \qquad \boldsymbol{L}_t[i,i] = 0 \;\; \forall i$$

$$\boldsymbol{L}_t[i,j] = \left(1 - \mathbf{w}_t^w[i] - \mathbf{w}_t^w[j]\right) \boldsymbol{L}_{t-1}[i,j] + \mathbf{w}_t^w[i]\mathbf{p}_{t-1}[j] \tag{11}$$

Forward and backward address distributions are given by $\mathbf{f}_t^i$ and $\mathbf{b}_t^i$:

$$\mathbf{f}_t^i = S\left(\boldsymbol{L}_t \mathbf{w}_{t-1}^{r,i}, s_t^{f,i}\right) \qquad \mathbf{b}_t^i = S\left(\boldsymbol{L}_t^{\mathsf{T}} \mathbf{w}_{t-1}^{r,i}, s_t^{b,i}\right) \qquad S(\mathbf{d}, s)_i = \frac{\left(\frac{\mathbf{d}_i + \epsilon}{\max\,(\mathbf{d}+\epsilon)}\right)^s}{\sum_j \left(\frac{\mathbf{d}_j + \epsilon}{\max\,(\mathbf{d}+\epsilon)}\right)^s} \tag{12}$$

Compare this to the $\mathbf{f}_t^i = \boldsymbol{L}_t \mathbf{w}_{t-1}^{r,i}$ and $\mathbf{b}_t^i = \boldsymbol{L}_t^{\mathsf{T}} \mathbf{w}_{t-1}^{r,i}$ of Graves et al. (2016).

The read address distribution is given by:

$$\mathbf{c}_t^{r,i} = C\left(\boldsymbol{M}_t, \mathbf{k}_t^{r,i}, \beta_t^{r,i}, \mathbf{m}_t^{r,i}\right) \tag{13}$$

$$\mathbf{w}_t^{r,i} = \pi_t^i[1]\mathbf{b}_t^i + \pi_t^i[2]\mathbf{c}_t^{r,i} + \pi_t^i[3]\mathbf{f}_t^i \tag{14}$$

Finally, memory is read, and the output is calculated:

$$\mathbf{y}_t = \mathbf{v}_t + \boldsymbol{W}_r \left[\mathbf{r}_t^1; ...; \mathbf{r}_t^R\right] \qquad \mathbf{r}_t^i = \boldsymbol{M}_t^{\mathsf{T}} \mathbf{w}_t^{r,i}$$

## B  HYPERPARAMETERS FOR THE EXPERIMENTS

**Copy Task.**  We use an LSTM controller with hidden size 32, memory of 16 words of length 16, 1 read head. $W$ is 8, with the 9th bit indicating the start of the repeat phase. $L$ is randomly chosen from range $[1,8]$, $N$ from range $[2,14]$. Batch size is 16.

**Associative Recall Task.**  We use a single-layer LSTM controller (size 128), memory of 64 cells of length 32, 1 read head. $W_b = 3$, $B \in [2,16]$, $W_b = 8$, batch size of 16.

**Key-Value Retrieval Task.**  We use a single-layer LSTM controller of size 32, 16 memory cells of length 32, 1 read head. $W = 8$, $L \in [2,16]$.

**bAbI.**  Our network has a single layer LSTM controller (hidden size of 256), 4 read heads, word length of 64, and 256 memory cells. Embedding size is $E = 256$, batch size is 2.

## C    ADDITIONAL BABI RESULTS

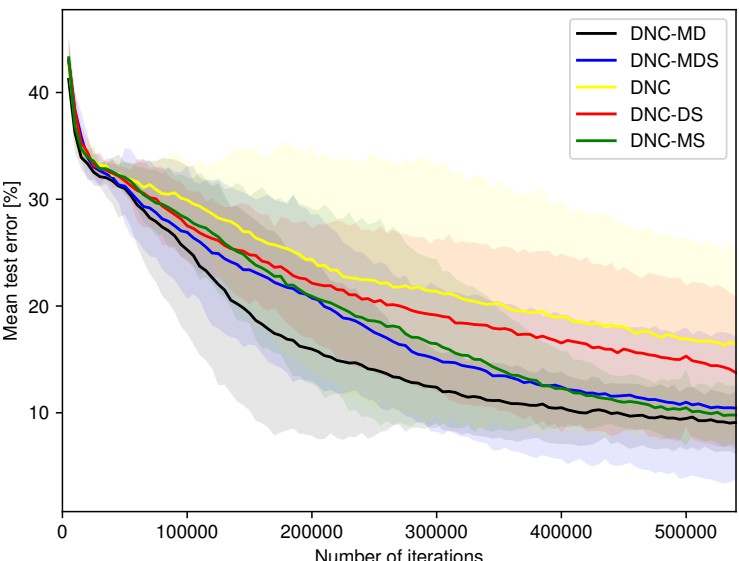

Figure 6: Mean test error of various models during the training. Shadowed area shows $\pm 2\sigma$.

