# OpenReview forum: "Improving Differentiable Neural Computers Through Memory Masking, De-allocation, and Link Distribution Sharpness Control"
_ICLR.cc/2019/Conference_

### Official Review · AnonReviewer1 · 2018-10-27
**Well written, has implications beyond DNC**

**Rating:** 7
**Confidence:** 5

**Review:**

The authors propose three improvements to the DNC model: masked attention, erasion of de-allocated elements, and sharpened temporal links --- and show that this allows the model to solve synthetic memory tasks faster and with better precision. They also show the model performs better on average on bAbI than the original DNC.

The negatives are that the paper does not really show this modified DNC can solve a task that the original DNC could not. As the authors also admit, there have been other DNC improvements that have had more dramatic improvements on bAbI.

I think the paper is particularly clearly written, and I would vote for it being accepted as it has implications beyond the DNC. The fact that masked attention works so much better than the standard cosine-weighted content-based attention is pretty interesting in itself. The insights (e.g. Figure 5) are interesting and show the study is not just trying to be a benchmark paper for some top-level results, but actually cares about understanding a problem and fixing it. Although most recent memory architectures do not seem to have incorporated the DNC's slightly complex memory de-allocation scheme, any resurgent work in this area would benefit from this study.

---

> ### Author Response · Authors · 2018-11-16
> **Reply to reviewer 1**
>
> Thank you for your thoughtful feedback!

---

### Official Review · AnonReviewer2 · 2018-11-02
**Promising modifications to the Differentiable Neural Computer (DNC) architecture, but needs stronger empirical evidence**

**Rating:** 7
**Confidence:** 5

**Review:**


Overview:
This paper proposes modifications to the original Differentiable Neural Computer architecture in three ways. First by introducing a masked content-based addressing which dynamically induces a key-value separation. Second, by modifying the de-allocation system by also multiplying the memory contents by a retention vector before an update. Finally, the authors propose a modification in the link distribution, through renormalization. They provide some theoretical motivation and empirical evidence that it helps avoiding memory aliasing.
The authors test their approach in the some algorithm task from the DNC paper (Copy, Associative Recall and Key-Value Retrieval), and also in the bAbi dataset.


Strengths: Overall I think the paper is well-written, and proposes simple adaptions to the DNC architecture which are theoretically grounded and could be effective for improving general performance. Although the experimental results seem promising when comparing the modified architecture to the original DNC, in my opinion there are a few fundamental problems in the empirical session (see weakness discussion bellow).

Weaknesses: Not all model modifications are studied in all the algorithmic tasks. For example, in the associative recall and key-value retrieval only DNC and DNC + masking are studied.

For the bAbi task, although there is a significant improvement (43%) in the mean error rate compared to the original DNC, it's important to note that performance in this task has improved a lot since the DNC paper was release. Since this is the only non-toy task in the paper, in my opinion, the authors have to discuss current SOTA on it, and have to cite, for example the universal transformer[1], entnet[2], relational nets [3], among others architectures that shown recent advances on this benchmark.
Moreover, the sparse DNC (Rae el at., 2016) is already a much better performant in this task. (mean error DNC: 16.7 \pm 7.6, DNC-MD (this paper) 9.5 \pm 1.6, sparse DNC 6.4 \pm 2.5). Although the authors mention in the conclusion that it's future work to merge their proposed changes into the sparse DNC, it is hard to know how relevant the improvements are, knowing that there are much better baselines for this task.
It would also be good if besides the mean error rates, they reported best runs chosen by performance on the validation task, and number of the tasks solve (with < 5% error) as it is standard in this dataset.


Smaller Notes.
1) In the abstract, I find the message for motivating the masking from the sentence  "content based look-up results... which is not present in the key and need to be retrieved."  hard to understand by itself. When I first read the abstract, I couldn't understand what the authors wanted to communicate with it. Later in 3.1 it became clear.

2) page 3, beta in that equation is not defined

3) First paragraph in page 5 uses definition of acronyms DNC-MS and DNC-MDS before they are defined.

4) Table 1 difference between DNC and DNC (DM) is not clear. I am assuming it's the numbers reported in the paper, vs the author's implementation?

5)In session 3.1-3.3, for completeness. I think it would be helpful to explicitly compare the equations from the original DNC paper with the new proposed ones.

--------------

Post rebuttal update: I think the authors have addressed my main concern points and I am updating my score accordingly.

---

> ### Author Response · Authors · 2018-11-16
> **Reply to reviewer 2**
>
> Thank you for your thoughtful and helpful comments.
>
> Following the suggestions, we added additional results for the associative recall task for many network variants. We also report mean and variance of losses for different seeds. This shows that masking improves performance on this task especially when combined with improved de-allocation, while sharpness enhancements negatively affect performance in this case. From the variance plots it can be seen that some seeds of DNC-M and DNC-MD converge significantly faster than plain DNC.
>
> In our experimental section, we added requested references to methods performing better on bAbI, and point out that our goal is not to beat SOTA on bAbI, but to exhibit and overcome drawbacks of DNC.
>
> Comparison to Sparse DNC is an interesting idea, and we are currently running experiments in this direction. We intend to make the results available in the near future.
>
> We are unable to provide a fair comparison for the lowest bAbi scores, having reported 8 seeds compared to the 20 seeds reported by Graves et al. Indeed, the high variance of DNC (Table 1) suggests that it may benefit a lot from exploring additional seeds.
>
> We incorporated all of the smaller notes, including a comparison to the original DNC equations in Appendix A.

---

> > ### Comment · AnonReviewer2 · 2018-11-17
> > **rebuttal**
> >
> > Thanks for addressing the main concerns of my review, I have updated my score accordingly.

---

### Official Review · AnonReviewer3 · 2018-11-08
**Solid improvements to DNC.**

**Rating:** 8
**Confidence:** 5

**Review:**

Summary:

This paper is built on the top of DNC model. Authors observe a list of issues with the DNC model: issues with deallocation scheme, issues with the blurring of forward and backward addressing, and issues in content-based addressing. Authors propose changes in the network architecture to solve all these three issues. With toy experiments, authors demonstrate the usefulness of the proposed modifications to DNC. The improvements are also seen in more realistic bAbI tasks.

Major Comments:

The paper is well written and easy to follow. The proposed improvements seem to result in very clear improvements. The proposed improvements also improve the convergence of the model. I do not have any major concerns about the paper. I think that contributions of the paper are good enough to accept the paper.

I also appreciate that the authors have submitted the code to reproduce the results.

I am curious to know if authors observe similar convergence gains in bAbI tasks as well. Can you please provide the mean learning curve for bAbI task for DNC vs proposed modifications?

---

> ### Author Response · Authors · 2018-11-16
> **Reply to reviewer 3**
>
> Thank you for your careful consideration and feedback. Following your request, we updated the paper to include mean learning curves for different models in Figure 6 in Appendix C. Our models converge faster than DNC. Some of them (especially DNC-MD) also have significantly lower variance than DNC.

---

### Author Response · Authors · 2018-11-16
**Update**

Following the suggestions of the reviewers, we updated our paper. We made the following changes:
    - Clarified the abstract
    - Added mean/std loss curves for the associative recall task for many models
    - Added mean/std error curves for the bAbI task in the appendix
    - Highlighted our modifications compared to DNC equations in Appendix A
    - Fixed missing definitions/variables/etc.

---

### Meta-Review · Area_Chair1 · 2018-12-10
**Good paper; solid improvements to DNC**

**Confidence:** 4
**Recommendation:** Accept (Poster)

**Metareview:**


pros:
- Identification of several interesting problems with the original DNC model: masked attention, erasion of de-allocated elements, and sharpened temporal links
- An improved architecture which addresses the issues and shows improved performance on synthetic memory tasks and bAbI over the original model
- Clear writing

cons:
- Does not really show this modified DNC can solve a task that the original DNC could not and the bAbI tasks are effectively solved anyway.  It is still not clear whether the DNC even with these improvements will have much impact beyond these toy tasks.

Overall the reviewers found this to be a solid paper with a useful analysis and I agree.  I recommend acceptance.